behaviour/theoretical biology/ecology

movement, patch dynamics, demography, ideal free distribution, model

**Author for correspondence:**
Jing Jiao
e-mail: jjiao3@utk.edu

# Mobility and its sensitivity to fitness differences determine consumer–resource distributions

Jing Jiao[1,2], Louise Riotte-Lambert[4], Sergei S. Pilyugin[3], Michael A. Gil[5] and Craig W. Osenberg[6]

[1]NIMBioS, University of Tennessee, Knoxville, TN, USA
[2]Department of Biology, and [3]Department of Mathematics, University of Florida, Gainesville, FL, USA
[4]Institute of Biodiversity, Animal Health and Comparative Medicine, University of Glasgow, Glasgow, UK
[5]Institute of Marine Sciences, University of California, NOAA Southwest Fisheries Science Center, Santa Cruz, CA, USA
[6]Odum School of Ecology, University of Georgia, Athens, GA, USA

JJ, 0000-0003-2666-6452; LR-L, 0000-0002-6715-9898;
SSP, 0000-0001-5014-2232; MAG, 0000-0002-0411-8378;
CWO, 0000-0003-1918-7904

An animal's movement rate (mobility) and its ability to perceive fitness gradients (fitness sensitivity) determine how well it can exploit resources. Previous models have examined mobility and fitness sensitivity separately and found that mobility, modelled as random movement, prevents animals from staying in high-quality patches, leading to a departure from an ideal free distribution (IFD). However, empirical work shows that animals with higher mobility can more effectively collect environmental information and better sense patch quality, especially when the environment is frequently changed by human activities. Here, we model, for the first time, this positive correlation between mobility and fitness sensitivity and measure its consequences for the populations of a consumer and its resource. In the absence of consumer demography, mobility alone had no effect on system equilibria, but a positive correlation between mobility and fitness sensitivity could produce an IFD. In the presence of consumer demography, lower levels of mobility prevented the system from approaching an IFD due to the mixing of consumers between patches. However, when positively correlated with fitness sensitivity, high mobility led to an IFD. Our study demonstrates that the expected covariation of animal movement attributes can drive broadly theorized consumer–resource patterns across space and time and could underlie the role of consumers in driving spatial heterogeneity in resource abundance.

# 1. Introduction

Animal movement is central to behavioural ecology, meta-population and meta-community dynamics, epidemiology and conservation ecology [1–4]. Most animals move across spatially heterogeneous environments, in which they select habitat patches by being either more attracted to, or retained for longer, in patches that provide higher fitness [5,6]. The dependency of movement on fitness differences across space, commonly studied as 'habitat choice' or 'habitat selection' in the movement ecology literature (e.g. [7]), is considered an evolutionary adaptation to heterogeneous environments [8,9] and can lead to an ideal free distribution (IFD), in which consumers distribute themselves among patches so that their fitnesses (often approximated by their consumption rates of resources) are equal across occupied patches [10–12].

Most models of fitness-directed movement have distinguished two components: the baseline movement rate (hereafter referred to as 'mobility') and the responses to spatial differences in fitness (i.e. habitat selection behaviour, hereafter referred to as 'fitness sensitivity'), which is, in part, determined by the animal's perceptual abilities [13]. Previous theoretical patch models (e.g. [14,15]) combined baseline mobility and fitness sensitivity to represent fitness-directed movement, by allowing animals to direct movement towards the 'better' patch(es) while retaining some degree of 'mistakes', due, for example, to imperfect perception [13,16]. However, those models were only evaluated by varying either mobility or fitness sensitivity [14,15]. They therefore essentially treated the two processes as independent of one another. A similar approach has been taken in non-patch models that have represented space continuously [17,18] and in individual-based models [19]. In these models, no matter how space is modelled, it was also typically assumed that the environment was spatially variable but temporally constant and that the animals were able to instantaneously perceive the local environment's conditions. Under those assumptions, the models showed that mobility, acting as random movement, prevented individuals from staying in the best patch(es), while fitness sensitivity enhanced the ability of individuals to find and stay in the most profitable patches. Therefore, in these models, mobility and fitness sensitivity had antagonistic effects on the system achieving an IFD (see [17,20]). This finding is consistent with other theoretical studies comparing the effects of random and fitness-directed movement on system dynamics (see [8,21,22]).

However, mobility and fitness sensitivity of movement are not likely to be independent. For example, marine placozoans with higher mobility are better able to sense and move towards food [23]. This could result from the positive relationship between mobility and information acquisition [24]. Many animals collect information about their environment while moving, a phenomenon commonly called 'habitat sampling' [13,16,25–27]. Moreover, natural environments change with time, due to both natural and anthropogenic influences (e.g. [28]). Therefore, higher mobility could increase information acquisition, shorten the time required to detect and exploit transient fitness differences [13,16] and, thus, potentially increase the fitness sensitivity of movement.

By ignoring the interdependency between mobility and the fitness sensitivity of movement, previous studies cannot measure how interactions between mobility and fitness sensitivity may shape distributions of animal populations. This could lead to the incomplete understanding, or even misinterpretation, of the effects of animal movement on system dynamics. For example, previous studies, which assumed that mobility and fitness sensitivity were independent, have postulated that an IFD is most likely when mobility is low and fitness sensitivity is high [17,20]. However, a positive correlation between mobility and fitness sensitivity could yield results that qualitatively change these expectations. To explore the conditions under which mobility and fitness sensitivity interactively affect consumer–resource distributions, here we compare the results when mobility and fitness sensitivity vary independently of one another (as in past analyses) with results when they covary.

We develop a simple two-patch consumer–resource model to investigate the effects of the positive correlation between mobility and the fitness sensitivity of movement on population dynamics and the spatial distributions of consumers and resources. We used both analytical and numerical methods to explore the isolated and combined effects of mobility and fitness sensitivity of movement on the equilibrium densities of consumers and resources at the local (within one patch) and regional (across both patches) scales. To separate the effects of behavioural processes from longer-term population dynamics, we first excluded consumer demography to examine patterns arising over short (i.e. behavioural) temporal scales and then included consumer demography to examine long-term patterns arising from both behavioural and demographic processes.

# 2. Model

We consider a consumer–resource system within a two-patch landscape that is heterogeneous in resource dynamics (see [29]). For simplicity, we consider two patches: a high-quality patch containing resources with a high growth rate and large carrying capacity, and a low-quality patch containing resources with a low growth rate and small carrying capacity. We assume that patches are equal in size and that each patch can support a local community of resources and consumers. Because consumers are often much more mobile than their resources (e.g. grazers versus grass, fishes versus zooplankton or snails versus barnacles; see [30–32]), and to facilitate analytical tractability, we assume that consumers can move but resources are sedentary. We quantify the fitness of the consumers as their local per-capita growth rate. We assume that resources grow logistically in the absence of consumers, consumers exhibit a Type I functional response and consumers do not exhibit interference competition (interference competition and saturating functional responses can produce confounding effects on the densities of consumers and resources; see [14]). These assumptions lead to the following equations:

$$\frac{dR_H}{dt} = r_H R_H \left(1 - \frac{R_H}{K_H}\right) - \alpha R_H C_H, \tag{2.1}$$

$$\frac{dR_L}{dt} = r_L R_L \left(1 - \frac{R_L}{K_L}\right) - \alpha R_L C_L, \tag{2.2}$$

$$\frac{dC_H}{dt} = p(c\alpha R_H - \mu)C_H + C_L Q_{HL} - C_H Q_{LH} \tag{2.3}$$

and

$$\frac{dC_L}{dt} = p(c\alpha R_L - \mu)C_L + C_H Q_{LH} - C_L Q_{HL}, \tag{2.4}$$

where the state variables, $R_i$ and $C_i$, are the densities of resources and consumers, respectively, in patch type $i$ (low-quality = L and high-quality = H), $r_i$ and $K_i$ are the growth rate and the carrying capacity of the resources in patch $i$, $\alpha$ is the attack rate of consumers on resources, $c$ is the consumer's conversion efficiency, $\mu$ is the consumer's mortality rate, $Q_{ij}$ is the movement rate of consumers from patch $j$ to $i$ (we assume that individuals are well mixed inside each patch) and $p$ is a parameter that indicates the inclusion ($p = 1$) or exclusion ($p = 0$) of consumer demography.

The fitness of consumers in patch $i$ ($w_i$) is equal to their per-capita growth rate:

$$w_i = c\alpha R_i - \mu. \tag{2.5}$$

To describe the movement rate from patch $j$ to $i$, we follow the general form used in previous theoretical investigations (e.g. [14,15,33]:

$$Q_{ij} = \beta e^{\lambda(w_i - w_j)}, \tag{2.6}$$

where $\beta$ is the consumer's mobility, $\lambda$ is the fitness sensitivity of movement and $w_i - w_j$ is the fitness difference between patch $i$ and $j$.

Using this model, we examined how the equilibria of consumers and resources change when $\beta$ and $\lambda$ vary independently of each other versus how they change when $\beta$ and $\lambda$ positively covary. For simplicity but without losing generality, we assumed that $\beta$ and $\lambda$ were linearly related, i.e. $\lambda = \gamma \beta$, where $\gamma > 0$. We restricted our analyses to parameter values that led to positive consumer and resource densities in both patches, e.g. we did not consider cases in which the consumer was unable to persist in the low-quality patch in the absence of migration. When analytical solutions were not possible, we conducted simulations in the R programming language [34]. Each simulation was run for 5000 time-steps, which was sufficient for the system to reach equilibrium (i.e. densities did not change with additional time-steps).

# 3. Results

## 3.1. Without consumer demography

In the absence of consumer demography ($p = 0$ in equations (2.3) and (2.4)), the model had a unique stable positive equilibrium (see electronic supplementary material, appendix 1), which occurred when

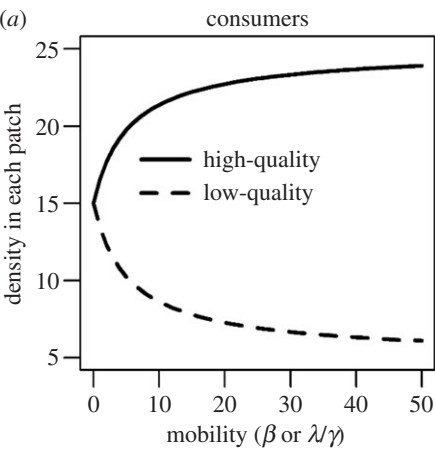
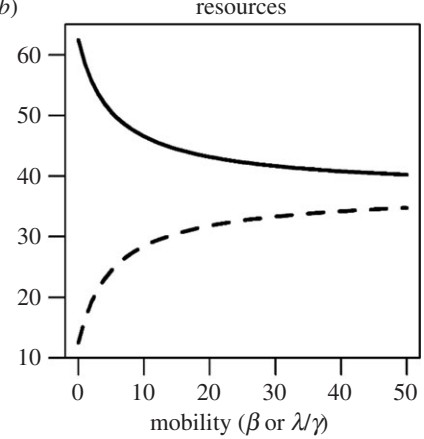

**Figure 1.** Relationships between equilibrium densities of consumers (*a*), resources (*b*) and mobility ($\beta$ or $\lambda/\gamma$) in the high- (solid line) or the low-quality patch (dashed line) in the absence of consumer demography ($p = 0$). The parameters are: $r_H = 2$, $r_L = 1$, $K_H = 100$, $K_L = 50$, $c = 0.05$, $\alpha = 0.05$, $\mu = 0.1$ and $C_T = 15$. These results do not directly depend on $\beta$ and the correlation between mobility and fitness sensitivity. The *x*-axis scale here is for mobility, but it indicates an increase in both mobility and fitness sensitivity because we assume that mobility and fitness sensitivity covary ($\lambda = \gamma\beta$ where $\gamma > 0$).

total migration rates into and out of each patch were equal ($C_L Q_{HL} = C_H Q_{LH}$):

$$\frac{C_L^*}{C_H^*} = e^{-2\lambda c \alpha [R_H^* - R_L^*]}, \tag{3.1}$$

$$R_H^* = K_H \left(1 - \frac{\alpha C_H^*}{r_H}\right), \tag{3.2}$$

and
$$R_L^* = K_L \left(1 - \frac{\alpha C_L^*}{r_L}\right). \tag{3.3}$$

Solutions for the equilibria (equations (3.1)–(3.3)) did not include the mobility term $\beta$, indicating that changing mobility alone (i.e. without a correlated change in fitness sensitivity) does not affect the equilibrial consumer–resource patterns. However, greater mobility led to faster convergence to the steady state (see electronic supplementary material, figure S1 in appendix 2). By contrast, solutions for the equilibria did depend on fitness sensitivity, $\lambda$ (via equation (3.1)). Therefore, we conducted extra analyses (in the absence of consumer demography) to explore how the equilibria changed as $\lambda$ increased (which yields identical equilibria as the situation in which $\beta$ and $\lambda$ are positively correlated).

In the absence of fitness sensitivity (i.e. $\lambda = 0$), consumers were equally distributed between the two patches ($C_H^* = C_L^* = C_T$, where $C_T$ is the mean consumer density; see the *y*-intercept in figure 1*a*), while the resources were more dense in the high-quality patch ($R_H^* > R_L^*$; compare the solid and the dashed lines in figure 1*b*).

As fitness sensitivity increased, consumer density increased in the high-quality patch but decreased in the low-quality patch at equilibrium (figure 1*a*), while resource density showed the opposite pattern (figure 1*b*; see also equations S16 and S18 in electronic supplementary material, appendix 3). Thus, the disparity of resource densities in the two patches decreased as fitness sensitivity increased, and eventually approached 0 as fitness sensitivity approached infinity. As a result, consumer fitness (per-capita growth rate; equation (2.5)) became equal in the two patches. In other words, with high fitness sensitivity, consumers distributed themselves among patches to achieve equal fitness, resulting in an IFD (see [10,18]).

Due to the absence of consumer demography, the average (system-wide) density of the consumer was constant across all parameter values (figure 2*a*). The resources, however, were dynamic, yet their average density was also invariant to changes in $\lambda$ so long as the density-dependent mortality of resources was equivalent in both patches ($r_H/K_H = r_L/K_L$), i.e. as the fitness sensitivity of the consumers' movement increased, the regional density of resources remained constant (the solid line in figure 2*b*; equation S21 in electronic supplementary material, appendix 4). If the density-dependent mortality of resources was larger in the high-quality patch (i.e. $r_H/K_H > r_L/K_L$; equation S22 in electronic supplementary

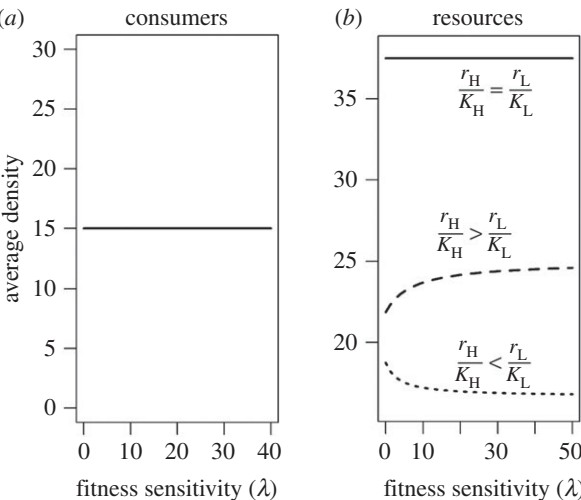

**Figure 2.** Relationship between average density of consumers (*a*) and resources (*b*) and fitness sensitivity ($\lambda$) in each patch in the absence of consumer demography ($p = 0$). In (*b*), parameters of the solid line are: $r_H = 2$, $r_L = 1$, $K_H = 100$ and $K_L = 50$; parameters of the dashed line are: $r_H = 2$, $r_L = 1$, $K_H = 50$; parameters of the dotted line are: $r_H = 1$, $r_L = 1$, $K_H = 100$, $K_L = 50$. Other parameters are: $c = 0.05$, $\alpha = 0.05$, $\mu = 0.1$, $C_T = 15$ and $\gamma = 1$.

material, appendix 4; see the dashed line in figure 2*b*), the movement of consumers from the low- to the high-quality patch reduced intraspecific competition in the resource and increased average resource density (see compensatory growth in [35]). Therefore, increasing fitness sensitivity reinforced the increase in regional resource density (equation S22 in electronic supplementary material, appendix 4 and the dashed line in figure 2*b*). Conversely, if $r_H/K_H < r_L/K_L$, intraspecific competition in the resource increased when consumers migrated from the low- to the high-quality patch, causing the regional resource density to decline (equation S23 in electronic supplementary material, appendix 4; see the dotted line in figure 2*b*).

## 3.2. With consumer demography

In the presence of consumer demography ($p = 1$ in equations (2.3) and (2.4)), the equilibria were determined by mobility, the fitness sensitivity of movement and the demographic parameters of consumers within each patch. Thus, in the presence of consumer demography, we considered all three movement scenarios: (i) fitness sensitivity ($\lambda$) was fixed but mobility ($\beta$) varied; (ii) mobility was fixed but fitness sensitivity varied, and (iii) fitness sensitivity and mobility covaried (i.e. $\lambda = \gamma \beta$, where $\gamma > 0$). In a few simple cases, we were able to provide analytical solutions, although in most cases, we relied on simulations. Here, we only considered the parameter sets that led to stable equilibria. It is possible that other parametrizations could produce oscillations (for details, see [33]), but this is beyond the scope of this paper.

### 3.2.1. Fitness sensitivity is fixed but mobility varies

To evaluate the independent effects of mobility, we fixed fitness sensitivity at three values ($\lambda = 0.1$, 10 or 1000) but varied mobility ($\beta$). In the absence of movement ($\beta = 0$), the two patches were not coupled, and the equilibria were determined by consumer demography alone:

$$R_H^* = R_L^* = R^* = \frac{\mu}{c\alpha}, \tag{3.4}$$

$$C_H^* = \frac{r_H}{\alpha}\left(1 - \frac{\mu}{c\alpha K_H}\right) \tag{3.5}$$

and
$$C_L^* = \frac{r_L}{\alpha}\left(1 - \frac{\mu}{c\alpha K_L}\right). \tag{3.6}$$

Thus, in the absence of movement, at equilibrium, consumer density was greater in the high-quality patch (due to $r_H > r_L$ and $K_H > K_L$: figure 3*a*,*b* and equations (3.5) and (3.6)), but resource density and

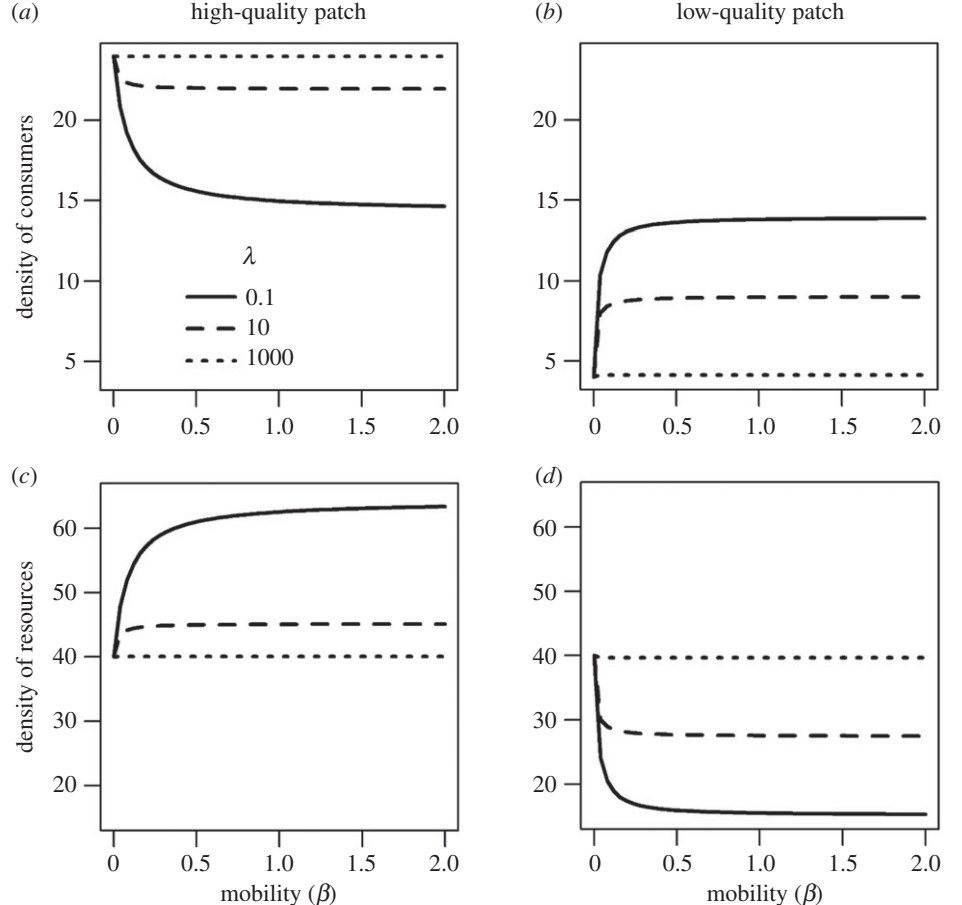

**Figure 3.** Relationship between equilibrium densities of consumers ($a,b$), resources ($c,d$) and mobility ($\beta$) in the high- ($a,c$) and the low-quality patch ($b,d$) in the presence of consumer demography ($p = 1$). Fitness sensitivity is set at three levels: $\lambda = 0.1$, 10 and 1000. Other parameters are: $r_H = 2$, $r_L = 1$, $K_H = 100$, $K_L = 50$, $c = 0.05$, $\alpha = 0.05$ and $\mu = 0.1$.

thus consumer fitness were equal in the two patches (figure 3$c,d$; see equation (2.5)). This is the classic result from the Lotka–Volterra predator–prey model, in which the resource equilibrium is set by the mortality rate of the predator (i.e. consumer), which is the same in both patches [36].

Because consumers tended to be denser in the high-quality patch, increasing mobility ($\beta$) moved consumers out of the high-quality patch and into the low-quality patch. As a result, increasing mobility ($\beta$) reduced the density disparity of consumers between the patches, although this was counteracted by the level of fitness sensitivity of movement: consumer density was more different between the patches when fitness sensitivity was larger (figure 3$a,b$). Increasing the mobility of consumers led to a larger difference of resource densities between the two patches, i.e. higher resource density in the high-quality patch but lower resource density in the low-quality patch (figure 3$c,d$). This difference in resource density was greater when fitness sensitivity was smaller. Similarly, the fitness of consumers in the two patches (equation (2.5)) became more disparate as mobility increased, mirroring the change in the resource density (figure 3$c,d$). This effect was reduced under larger fitness sensitivity because fitness sensitivity produced the opposite effect on the movement of consumers, i.e. more consumers moved from the low- to the high-quality patch. When fitness sensitivity was very large (e.g. $\lambda = 1000$), changing mobility had little effect on consumers or resources, i.e. the equilibrium densities of consumers and resources resembled those that occurred when consumers were immobile and could only respond demographically (see the dotted lines in figure 3).

### 3.2.2. Mobility is fixed but fitness sensitivity varies

For these analyses, we fixed mobility at three levels ($\beta = 0.01$, 0.1 or 1) but varied fitness sensitivity ($\lambda$). In the absence of fitness sensitivity ($\lambda = 0$), consumers exhibited purely random movement (i.e. all individuals had the same probability of emigration and no matter the density of resources in a patch)

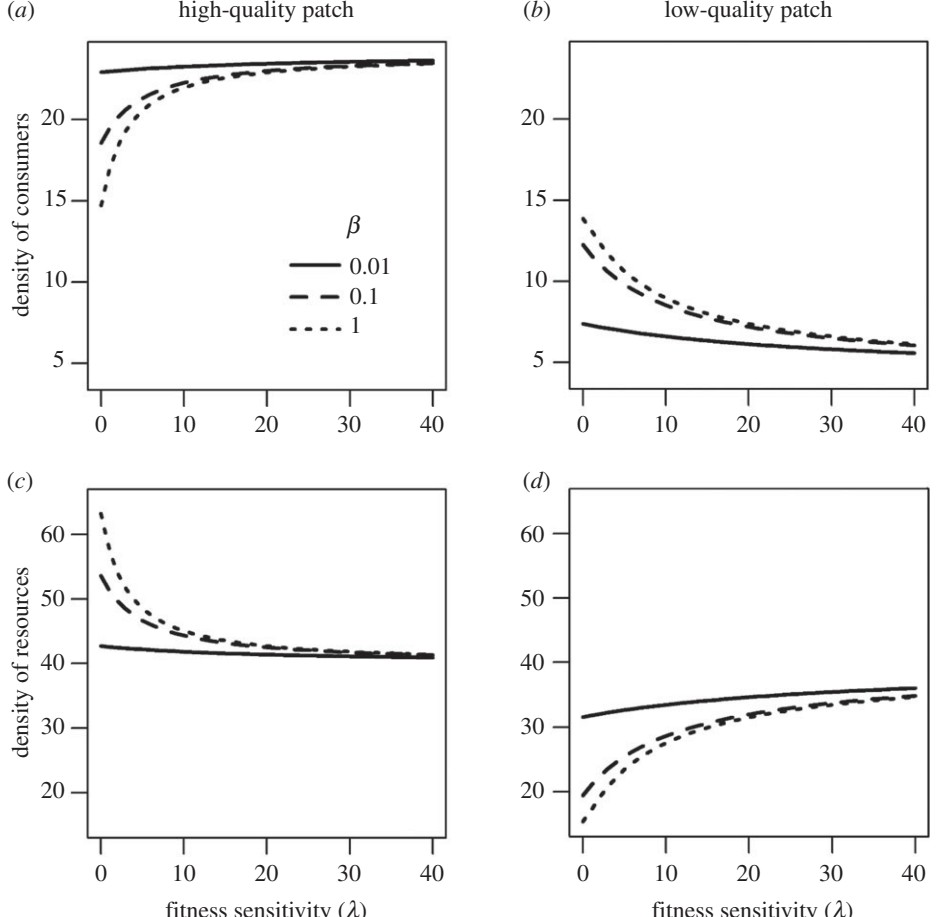

**Figure 4.** Relationship between equilibrium densities of consumers (a,b), resources (c,d) and fitness sensitivity ($\lambda$) in the high- (a,c) and the low-quality patch (b,d) in the presence of consumer demography ($p = 1$). Mobility is set at three levels: $\beta = 0.01$, 0.1 and 1. Other parameters are: $r_H = 2$, $r_L = 1$, $K_H = 100$, $K_L = 50$, $c = 0.05$, $\alpha = 0.05$ and $\mu = 0.1$.

and had higher birth rates in the high-quality patch and thus achieved higher densities. Because of this density difference, more consumers moved from the high- to the low-quality patch, and mobility controlled the rate of this movement (larger $\beta$ homogenized the distribution of consumers between patches; see the nearly equal intercepts of the dotted lines in figure 4a,b). This spilling of consumers out of the high-quality patch created a disparity in resource densities between the two patches (compare intercepts in figure 4c,d).

The fitness sensitivity of movement counteracted the homogenizing effect of mobility on consumer density. As a result, as we increased fitness sensitivity, consumer density increased in the high-quality patch and decreased in the low-quality patch (figure 4a,b). Resource density became more similar in the two patches (figure 4c,d). These effects of fitness sensitivity were more pronounced when animals were more mobile (compare the trends of the three lines in figure 4).

### 3.2.3. Fitness sensitivity and mobility vary together

To explore the effects of covariance in fitness sensitivity and mobility, we rewrote equation (2.6) by replacing $\lambda$ with $\gamma\beta$. Thus, although we functionally varied $\beta$ to conduct our simulations, this effect was equivalent to simultaneously changing both $\lambda$ and $\beta$. We examined the effect of increasing movement (i.e. mobility) at three different levels of fitness sensitivity relative to mobility, $\gamma$ (=$\lambda/\beta$; 0.1, 1 and 10).

As we noted above, in the absence of any movement ($\lambda=\beta=0$), equilibria were determined by consumer demography alone: consumer density was greater in the high-quality patch, but resources (and thus consumer fitness) were equivalent in the two patches (see equations (2.5) and (3.4)–(3.6) and y-intercepts in figure 5). As movement (i.e. both mobility and fitness sensitivity) slightly increased, the disparity in consumer density decreased, but the disparity in resource density increased (figure 5).

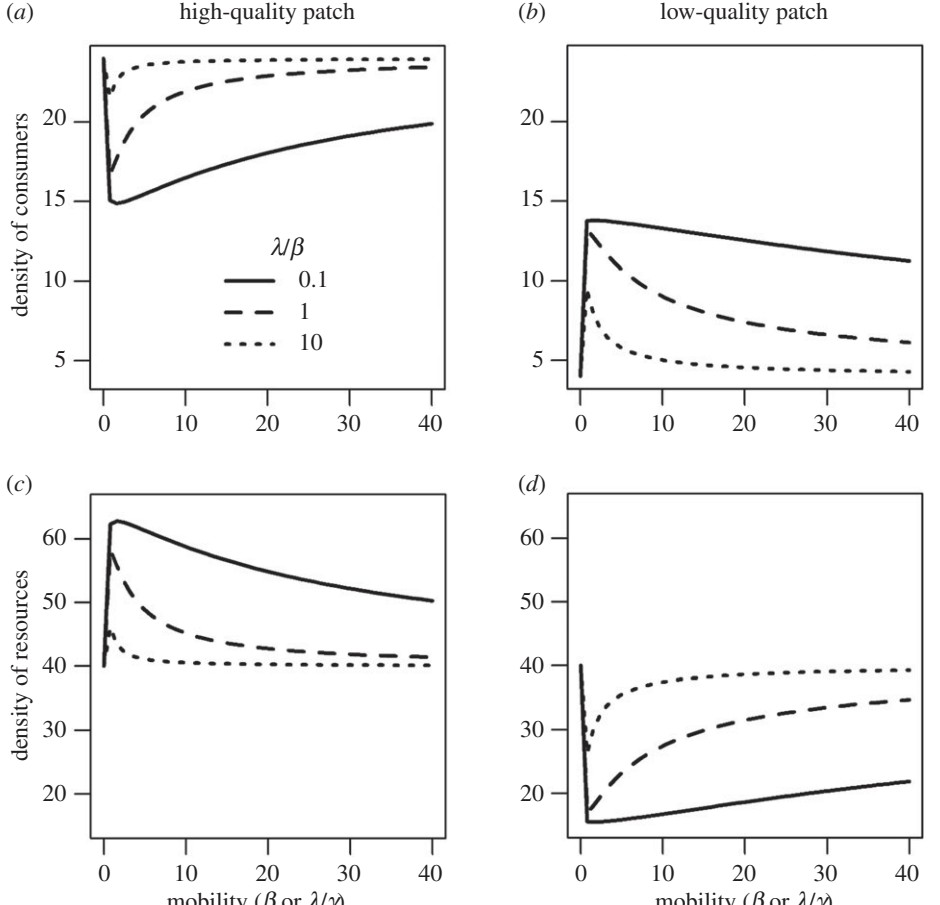

**Figure 5.** Relationship between equilibrium densities of consumers ($a,b$), resources ($c,d$) and mobility ($\beta$ or $\lambda/\gamma$) under three ratios of fitness sensitivity versus mobility in the high- ($a,c$) and the low-quality patch ($b,d$) in the presence of consumer demography ($p = 1$). The x-axis scale is for mobility, but it indicates an increase in both mobility and fitness sensitivity because we assume that mobility and fitness sensitivity covary ($\lambda = \gamma\beta$) at $\gamma = 0.1$, 1 and 10. Other parameters are: $r_H = 2$, $r_L = 1$, $K_H = 100$, $K_L = 50$, $c = 0.05$, $\alpha = 0.05$ and $\mu = 0.1$.

This pattern was similar to the scenario in which fitness sensitivity was fixed but only mobility varied (figure 3). However, as movement increased further (e.g. when $\beta > 2$), the opposite trend occurred: the disparity in consumer density increased while the disparity in resource density decreased. This trend was more obvious when fitness sensitivity was relatively large (indicated by the larger values of $\gamma = \lambda/\beta$; compare the three lines in figure 5). The above trends resulted in a unimodal relationship between consumer density and movement (figure 5). Consumers and resources exhibited opposite patterns at equilibrium: higher consumer density led to lower resource density in either the high- or low-quality patch (compare the three lines in ($a,b$) and ($c,d$) in figure 5). These humped relationships suggest that random movement dominated the system when consumers were less mobile, while fitness-directed movement dominated the system when consumers were more mobile.

## 4. Discussion

Movement can influence species interactions and distributions across heterogeneous landscapes [37–44]. Previous studies of fitness-directed movement used a modelling framework similar to ours (i.e. equation (2.6)), but examined effects of mobility and fitness sensitivity separately. These studies showed that the two processes have opposing effects on achieving an IFD: increasing fitness sensitivity facilitated the formation of an IFD (via increased movement into patches offering higher fitness), but increasing mobility prevented it (by increased mixing among patches) (see figure 3; also [17,20,22]). By contrast, our study demonstrated a three-way interaction between mobility, fitness sensitivity and consumer demography. In the absence of consumer demography, only fitness sensitivity influenced

consumer–resource patterns: IFD was produced only under high rates of fitness sensitivity. Mobility can thus only influence IFD through a correlation with fitness sensitivity (figure 1). However, when consumers were demographically dynamic, consumer demography alone (without any movement) produced IFD-like patterns (figures 3–5; see [20]). In the presence of movement with uncorrelated mobility and fitness sensitivity, fitness sensitivity facilitated, but mobility prevented, the achievement of IFD (figure 3), which is consistent with previous theoretical work [17,20,22]. When mobility and fitness sensitivity were positively correlated, the relative role of their synergistic and antagonistic effects depended on the overall levels of movement (figure 5). At low movement rates, increasing movement disrupted the demographically driven IFD—the role of mobility dominated these effects of movement; however, when at higher movement rates, increasing movement caused the system to revert to IFD (see unimodal trend in figure 5). These results highlight the joint influence of consumer demography and movement on species interactions [45,46] and, in contrast to previous work [17,20], demonstrate how high mobility could *facilitate* an IFD.

Understanding the effects of a correlation between mobility and the fitness sensitivity of movement could be fundamental to reaching a unified understanding of the propensity of a system to reach an IFD, a pattern commonly observed in various natural ecosystems with highly mobile consumers and sessile resources [47,48]. The positive correlation between mobility and the fitness sensitivity of movement could be explained by the joint evolution of an animal's movement and sensory and cognitive capacities that enhance the ability to perceive and respond to environmental information [13,49] and by the fact that animals that are more mobile encounter more opportunities to collect information about how the environment varies over space [16]. Consistent with this idea, it has been shown that less mobile species often perform random movement and have limited exploration capability (e.g. deposit-feeding invertebrates fail to forage in food-rich areas; [50]), while more mobile individuals usually exhibit higher fitness sensitivity [23]. Moreover, mobility and fitness sensitivity are often indirectly linked through body size and allometric correlations. Indeed, many behavioural and physiological traits, such as the space used by animals [51], dispersal distance [52] and perceptual range [53], are positively correlated with body size [54].

However, some taxa rely heavily upon social cues to supplement the personal information that they collect directly from the environment [26,55,56]. For these individuals, fitness sensitivity may not be as strongly correlated with mobility because information about fitness differences can be acquired vicariously, and from greater distances than direct, personal observations of the environment [56]. Moreover, at the behavioural scale, some animals might experience a speed–accuracy trade-off: the perceptual capacity of individuals could decrease as their movement speed increases [16]. This trade-off would drive a negative relationship between movement speed and the quality of information collected while moving [16]. Our model can be easily adapted to reflect these additional layers of complexity by modifying the relationship between mobility ($\beta$) and fitness sensitivity ($\lambda$). Furthermore, contrary to our assumption that all consumers have the same mobility and fitness-directed movement, individuals in a given population can show consistent differences in their mobility, as well as their perceptual abilities. Intraspecific heterogeneities may affect the dynamics presented here but are outside the scope of our mean-field approach. Future studies could extend our model to explore these effects. Our model could also be extended to directly include temporal variation, the timescale of environmental change and the magnitude of spatial heterogeneities, each of which may interact with animal movement to affect consumer–resource patterns. We, however, suspect that considering more complex landscapes (i.e. spatially explicit landscapes, with more than two patch types) would not significantly alter the qualitative insights about the interactive effects of mobility, fitness sensitivity and consumer demography on consumer–resource spatial patterns. Our model also ignored costs of travel and delays in the movement of consumers among patches, which could alter predictions of some patch-based foraging models (e.g. [57]) or cause cyclical dynamics [58]. We anticipate that these effects would reduce the overall movement of consumers (increasing their retention in resident patch(es)) but would not qualitatively alter the relative distribution of consumers among patches (see [59]). Given that consumer demography itself can produce IFD-like patterns (figure 5), we suspect that the incorporation of consumer demography should tend to lessen any additional departures from IFD arising from travel costs or delayed movement responses. Evaluating these conjectures with simulations is worthy of future investigation.

In summary, our study highlights a new unified perspective of animal movement in the context of consumer–resource systems and provides a more flexible approach that considers both short- and long-term dynamics. Our study thus provides a foundation for the understanding of consumer–resource relationships across spatially heterogeneous landscapes, which are ubiquitous in natural

systems. Furthermore, by explicitly considering the interplay between animal movement and consumer demography with respect to consumer–resource interactions, our study contributes to a growing body of theoretical literature on the effects of animal movement on higher-level ecological dynamics [27,60,61]. Such theoretical advances are needed to inform a rapidly growing body of empirical work on animal movement ecology, aided by a surge in technological advances in animal tracking [62–64].

Data accessibility. The R code used to generate all results in this paper is available at: https://doi.org/10.6084/m9.figshare.c.5015324.

Authors' contributions. J.J. conceived of the original idea. J.J., L.R.-L. and S.P. developed the model. L.R.-L., M.G. and C.O. contributed to the biological interpretation of the results. All authors contributed to the writing of the manuscript.

Competing interests. The authors declare no competing interests.

Funding. This work was funded by the NSF (via OCE-1130359, DMS-1411853, the QSE3 IGERT Program: DGE-0801544, and a Postdoctoral Research Fellowship awarded to M.G.). J.J. was also funded by the China Scholarship Council (CSC). L.R.-L. was funded by a Newton International Fellowship from the Royal Society (grant no. NF161261) and a Marie Skłodowska-Curie Individual Fellowship from the EU's Horizon 2020 Research and Innovation Program (grant no. 794760).

Acknowledgements. We thank Robert Holt for early discussion of the model structure and the helpful comments from Nina Fefferman's lab. We also thank several anonymous reviewers for discussion and comments that significantly improved this paper.

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
