## [Reviewer comments · Royal Society Open Science]

Review History

RSOS-200247.R0 (Original submission)

Review form: Reviewer 1 (Darcy Visscher)

Is the manuscript scientifically sound in its present form?

Yes

Are the interpretations and conclusions justified by the results?

Yes

Is the language acceptable?

Yes

Do you have any ethical concerns with this paper?

No

Have you any concerns about statistical analyses in this paper?

No

Recommendation?

Accept with minor revision (please list in comments)

Comments to the Author(s)

In the manuscript entitled “Mobility and its sensitivity to fitness differences determine consumer-resource distributions”, the authors detail a simple 2 patch model where the distribution of consumers is determined by patch quality (resources), as well as the mobility, and fitness sensitivity of the consumers. The authors then explore the dynamics of the system they have created via a set of sensitivity analyses. In particular they allow both mobility and fitness sensitivity to covary.

I think that the manuscript is very well written and raises some important points for readers. Its strength is the simplicity of the system and the extrapolations which can be done based it.

The manuscript admits to some of the limitations of being a “mean field approach” (Eulerian) but I think it would be good for the authors to couch their study within the literature employing individual based models (Lagrangian). I admit to my bias coming from this world, but I see some interesting connections and also terminological differences that would make this a useful exercise. For instance, “fitness sensitivity”, the perception and biased movement along a gradient has been well studied, analogously, in the movement literature as sampling (the authors note this in their discussion on information) and memory. In particular, it has been noted that memory processes analogous to this paper’s fitness sensitivity may limit the spread of an individual governed by an underlying random walk, allowing a home range to form and the forager to be most efficient in depletable heterogeneous landscapes. While I realize that the authors can’t make all the connections to this vast literature, I think the ability of the authors to speculate out to a more complex landscape (consisting of more than 2 patches) would increase the readership and scope of the manuscript. In fact, they allude to this on line 285, and I want to hear how they thought this would work.

Like many models of this nature, we often assume that there is no cost with movement and movement occurs instantaneously. Would the authors speculate as to what may happen if this was included? We have numerous model examples how the marginal value theorem impacts the IFD, could this model add to that understanding?

Some minor comments:

The literature cited section has numerous instances with the full journal names are not capitalized and a few instances of title vs. sentence case being used for the article title. I suspect this ought to be standardized.

The x-axis label for figure 1 & 5 with mobility or fitness sensitivity isn’t clear. I also wonder about the y-axis of “density of”, as the patches have no dimension and are identical (except in quality) isn’t this just number of consumers and resources?

Line 63 – this sentence is long and could be cleaned up a bit to make the idea clearer.

Review form: Reviewer 2

Is the manuscript scientifically sound in its present form?

Yes

Are the interpretations and conclusions justified by the results?

Yes

Is the language acceptable?

Yes

Do you have any ethical concerns with this paper?

No

Have you any concerns about statistical analyses in this paper?

No

Recommendation?

Major revision is needed (please make suggestions in comments)

Comments to the Author(s)

This paper investigates the occurrence of the Ideal Free Distribution in a two-patch consumer-resource model under varying scenarios including and excluding consumer dynamics.

Interestingly, the paper proves that whereas previous research states that IFD will most likely occur when movement is low and fitness sensitivity is high, IFD can be promoted by higher movement, in case it is assumed to be positively correlated with fitness sensitivity. Overall this paper is well written and the research is interesting, a nice contribution to the understanding of spatial dynamics and the field of movement ecology.

However, I still have some suggestions to improve the scope and clarity of this paper.

1. First of all, I would advice to broaden the introduction by including the term habitat choice. This might be a synonym to fitness sensitivity and an often used term in the field of movement ecology. Still, it is nowhere mentioned in the paper at the moment. Additionally, it would be better to not include a formula in the introduction. This gives the impression that this paper is only valid for researchers applying a similar formula in an analytical paper. It would be better to rewrite this paragraph in general terms, avoiding the use of this formula. It is already mentioned in the Method section and that should suffice. Moreover, the reference list should be extended with papers studying habitat choice and/or IFD in IBM models. There are many researchers applying IBM's to investigate movement ecology and you can appeal them by also referring to IBM studies (e.g. Mortier F., Jacob S., Vandegehuchte M.L. & Bonte D. (2019) Habitat choice stabilizes metapopulation dynamics by enabling ecological specialisation. *Oikos*). Hence, try to avoid the impression that this paper is written strictly for researchers working with analytical or numerical models.

2. For clarity, it would be better to

(i) Clearly state the main features or a clear definition of an Ideal Free Distribution in the introduction. For some readers this might be general knowledge. But for others, this might not be the case.

(ii) Not refer to 'demography' but 'consumer demography' instead. Demography can refer to both the resource and the consumer whereas you only want to refer to consumer demography. Demography of the resource is always included in all simulations. (lines 241, 242, 253, 292,...)

3. When discussing a possible positive correlation between fitness sensitivity and movement, it might be useful to mention the possible indirect link with body size. According to allometric theory, body size is positively linked to many features of an individual (among which movement speed and perceptual range). You could extend your discussion by mentioning this (reference: Peters 1983).

Minor remarks:

30: add a reference

79: 'a two-patch landscape' instead of 'a landscape that is patchy'

130: change 'We therefore focus our subsequent analyses' to 'Therefore, extra analyses were conducted to'

133: Beta and lambda are correlated in this section? Clearly state this.

179: add a reference

243: 'through a possible correlation with fitness' instead of 'through fitness'

279: Shorten the sentence which starts here, it is too long.

284: I would refer to temporal variation instead of environmental change as the former is more general.

Decision letter (RSOS-200247.R0)

06-Apr-2020

Dear Dr JIAO,

On behalf of the Editors, I am pleased to inform you that your Manuscript RSOS-200247 entitled "Mobility and its sensitivity to fitness differences determine consumer-resource distributions" has been accepted for publication in Royal Society Open Science subject to minor revision in accordance with the referee suggestions. Please find the referees' comments at the end of this email.

The reviewers and handling editors have recommended publication, but also suggest some minor revisions to your manuscript. Therefore, I invite you to respond to the comments and revise your manuscript.

- Ethics statement

- Data accessibility

<http://datadryad.org/submit?journalID=RSOS&manu=RSOS-200247>

- Competing interests

- Authors' contributions

- Acknowledgements

- Funding statement

Because the schedule for publication is very tight, it is a condition of publication that you submit the revised version of your manuscript before 15-Apr-2020. Please note that the revision deadline will expire at 00.00am on this date. If you do not think you will be able to meet this date please let me know immediately.

- 1) A text file of the manuscript (tex, txt, rtf, docx or doc), references, tables (including captions) and figure captions. Do not upload a PDF as your "Main Document";
- 2) A separate electronic file of each figure (EPS or print-quality PDF preferred (either format should be produced directly from original creation package), or original software format);
- 3) Included a 100 word media summary of your paper when requested at submission. Please ensure you have entered correct contact details (email, institution and telephone) in your user account;
- 4) Included the raw data to support the claims made in your paper. You can either include your data as electronic supplementary material or upload to a repository and include the relevant doi within your manuscript. Make sure it is clear in your data accessibility statement how the data can be accessed;

5) All supplementary materials accompanying an accepted article will be treated as in their final form. Note that the Royal Society will neither edit nor typeset supplementary material and it will be hosted as provided. Please ensure that the supplementary material includes the paper details where possible (authors, article title, journal name).

If your manuscript is newly submitted and subsequently accepted for publication, you will be asked to pay the article processing charge, unless you request a waiver and this is approved by Royal Society Publishing. You can find out more about the charges at <https://royalsocietypublishing.org/rsos/charges>. Should you have any queries, please contact openscience@royalsociety.org.

Kind regards,
Lianne Parkhouse
Editorial Coordinator
Royal Society Open Science
openscience@royalsociety.org

on behalf of Professor Len Thomas (Associate Editor) and Kevin Padian (Subject Editor)
openscience@royalsociety.org

Associate Editor Comments to Author (Professor Len Thomas):

Thank-you for your submission, and for revising your code to improve it. We have now received two reviews of your work, and both authors are broadly positive. However, both made helpful editorial recommendations that will improve the scope and presentation. Neither suggest any re-analysis, so I am recommending your manuscript be accepted pending minor revisions. In making your revisions please take careful account of every comment and suggestion made by the reviewers and either make appropriate changes in your paper or justify why you have not done so in your cover letter. By my reading, all reviewer suggestions seem sensible, so making appropriate changes should be your first choice. I look forward to seeing your revised submission.

Reviewer comments to Author:

Reviewer: 1
Comments to the Author(s)

In the manuscript entitled "Mobility and its sensitivity to fitness differences determine consumer-

resource distributions”, the authors detail a simple 2 patch model where the distribution of consumers is determined by patch quality (resources), as well as the mobility, and fitness sensitivity of the consumers. The authors then explore the dynamics of the system they have created via a set of sensitivity analyses. In particular they allow both mobility and fitness sensitivity to covary.

I think that the manuscript is very well written and raises some important points for readers. Its strength is the simplicity of the system and the extrapolations which can be done based it.

The manuscript admits to some of the limitations of being a “mean field approach” (Eulerian) but I think it would be good for the authors to couch their study within the literature employing individual based models (Lagrangian). I admit to my bias coming from this world, but I see some interesting connections and also terminological differences that would make this a useful exercise. For instance, “fitness sensitivity”, the perception and biased movement along a gradient has been well studied, analogously, in the movement literature as sampling (the authors note this in their discussion on information) and memory. In particular, it has been noted that memory processes analogous to this paper’s fitness sensitivity may limit the spread of an individual governed by an underlying random walk, allowing a home range to form and the forager to be most efficient in depletable heterogeneous landscapes. While I realize that the authors can’t make all the connections to this vast literature, I think the ability of the authors to speculate out to a more complex landscape (consisting of more than 2 patches) would increase the readership and scope of the manuscript. In fact, they allude to this on line 285, and I want to hear how they thought this would work.

Like many models of this nature, we often assume that there is no cost with movement and movement occurs instantaneously. Would the authors speculate as to what may happen if this was included? We have numerous model examples how the marginal value theorem impacts the IFD, could this model add to that understanding?

Some minor comments:

The literature cited section has numerous instances with the full journal names are not capitalized and a few instances of title vs. sentence case being used for the article title. I suspect this ought to be standardized.

The x-axis label for figure 1 & 5 with mobility or fitness sensitivity isn’t clear. I also wonder about the y-axis of “density of”, as the patches have no dimension and are identical (except in quality) isn’t this just number of consumers and resources?

Line 63 – this sentence is long and could be cleaned up a bit to make the idea clearer.

Reviewer: 2

Comments to the Author(s)

This paper investigates the occurrence of the Ideal Free Distribution in a two-patch consumer-resource model under varying scenarios including and excluding consumer dynamics. Interestingly, the paper proves that whereas previous research states that IFD will most likely occur when movement is low and fitness sensitivity is high, IFD can be promoted by higher movement, in case it is assumed to be positively correlated with fitness sensitivity. Overall this paper is well written and the research is interesting, a nice contribution to the understanding of spatial dynamics and the field of movement ecology.

However, I still have some suggestions to improve the scope and clarity of this paper.

1. First of all, I would advice to broaden the introduction by including the term habitat choice. This might be a synonym to fitness sensitivity and an often used term in the field of movement

ecology. Still, it is nowhere mentioned in the paper at the moment. Additionally, it would be better to not include a formula in the introduction. This gives the impression that this paper is only valid for researchers applying a similar formula in an analytical paper. It would be better to rewrite this paragraph in general terms, avoiding the use of this formula. It is already mentioned in the Method section and that should suffice. Moreover, the reference list should be extended with papers studying habitat choice and/or IFD in IBM models. There are many researchers applying IBM's to investigate movement ecology and you can appeal them by also referring to IBM studies (e.g. Mortier F., Jacob S., Vandegehuchte M.L. & Bonte D. (2019) Habitat choice stabilizes metapopulation dynamics by enabling ecological specialisation. *Oikos*). Hence, try to avoid the impression that this paper is written strictly for researchers working with analytical or numerical models.

2. For clarity, it would be better to

(i) Clearly state the main features or a clear definition of an Ideal Free Distribution in the introduction. For some readers this might be general knowledge. But for others, this might not be the case.

(ii) Not refer to 'demography' but 'consumer demography' instead. Demography can refer to both the resource and the consumer whereas you only want to refer to consumer demography. Demography of the resource is always included in all simulations. (lines 241, 242, 253, 292,...)

3. When discussing a possible positive correlation between fitness sensitivity and movement, it might be useful to mention the possible indirect link with body size. According to allometric theory, body size is positively linked to many features of an individual (among which movement speed and perceptual range). You could extend your discussion by mentioning this (reference: Peters 1983).

Minor remarks:

30: add a reference

79: 'a two-patch landscape' instead of 'a landscape that is patchy'

130: change 'We therefore focus our subsequent analyses' to 'Therefore, extra analyses were conducted to'

133: Beta and lambda are correlated in this section? Clearly state this.

179: add a reference

243: 'through a possible correlation with fitness' instead of 'through fitness'

279: Shorten the sentence which starts here, it is too long.

284: I would refer to temporal variation instead of environmental change as the former is more general.

Author's Response to Decision Letter for (RSOS-200247.R0)

See Appendix A.

Decision letter (RSOS-200247.R1)

12-May-2020

Dear Dr JIAO:

On behalf of the Editors, I am pleased to inform you that your Manuscript RSOS-200247.R1 entitled "Mobility and its sensitivity to fitness differences determine consumer-resource distributions" has been accepted for publication in Royal Society Open Science subject to minor

revision in accordance with the referee suggestions. Please find the referees' comments at the end of this email.

The reviewers and Subject Editor have recommended publication, but also suggest some minor revisions to your manuscript. Therefore, I invite you to respond to the comments and revise your manuscript.

- Ethics statement

- Data accessibility

If you wish to submit your supporting data or code to Dryad (<http://datadryad.org/>), or modify your current submission to dryad, please use the following link:
<http://datadryad.org/submit?journalID=RSOS&manu=RSOS-200247.R1>

- Competing interests

- Authors' contributions

- Acknowledgements

- Funding statement

Because the schedule for publication is very tight, it is a condition of publication that you submit the revised version of your manuscript before 21-May-2020. Please note that the revision deadline will expire at 00.00am on this date. If you do not think you will be able to meet this date please let me know immediately.

on behalf of Professor Len Thomas (Associate Editor) and Kevin Padian (Subject Editor)
openscience@royalsociety.org

Associate Editor Comments to Author (Professor Len Thomas):

Thank-you for your re-submission, and for taking account of the reviewers' comments. The revisions are largely acceptable, and I am on the point of accepting the paper; however I do not believe you have dealt with Reviewer 1 comment 4 adequately. For one thing, your edited text (lines 287-290 without tracked changes, sentence starting "Our model...") does not form a grammatically correct or comprehensible (to me) sentence. Secondly, the reviewer asked you to speculate as to what would happen if you included a cost to movement and movement taking non-negligible time. You addressed the former obliquely and the latter not at all. Please can you make more of an effort to engage fully with relaxation of both assumptions and provide justification for your speculations.

Reviewer comments to Author:

Author's Response to Decision Letter for (RSOS-200247.R1)

See Appendix B.

Decision letter (RSOS-200247.R2)

27-May-2020

Dear Dr JIAO,

It is a pleasure to accept your manuscript entitled "Mobility and its sensitivity to fitness differences determine consumer-resource distributions" in its current form for publication in Royal Society Open Science.

on behalf of Professor Len Thomas (Associate Editor) and Kevin Padian (Subject Editor)
openscience@royalsociety.org

Appendix A

Dear Professor Thomas,

We thank you for the provisional acceptance of our manuscript "Mobility and its sensitivity to fitness differences determine consumer-resource distributions" for publication in *Royal Society Open Science*. We also thank the referees for their useful comments that have helped us extend the scope of our study and better convey our messages. We have revised our manuscript accordingly and are here resubmitting the revised version. Below, we respond in detail to the points raised by the referees. For clarity, we have extracted and numbered their comments and present them in *italics*; our responses follow and are indented). All co-authors have reviewed and approved the revised manuscript.

Sincerely,

Jing Jiao, on behalf of all authors.

Research Associate
NIMBioS
University of Tennessee

Associate Editor Comments to Author (Professor Len Thomas):

Thank-you for your submission, and for revising your code to improve it. We have now received two reviews of your work, and both authors are broadly positive. However, both made helpful editorial recommendations that will improve the scope and presentation. Neither suggest any re-analysis, so I am recommending your manuscript be accepted pending minor revisions. In making your revisions please take careful account of every comment and suggestion made by the reviewers and either make appropriate changes in your paper or justify why you have not done so in your cover letter. By my reading, all reviewer suggestions seem sensible, so making appropriate changes should be your first choice. I look forward to seeing your revised submission.

Thank you. We have responded to the comments of the reviewers and agree that they have helped improve the manuscript. In particular, we have: accepted all minor editorial changes suggested by the reviewers, modified the text to clarify portions of the manuscript identified by the reviewers as problematic, and modified Figures 1 and 5 to improve the meaning of the x-axis, as the reviewers' feedback.

Comments from Referee 1:

1. I think that the manuscript is very well written and raises some important points for readers. Its strength is the simplicity of the system and the extrapolations which can be done based it.

We appreciate the support from Referee 1.

2. The manuscript admits to some of the limitations of being a “mean field approach” (Eulerian) but I think it would be good for the authors to couch their study within the literature employing individual based models (Lagrangian). I admit to my bias coming from this world, but I see some interesting connections and also terminological differences that would make this a useful exercise. For instance, “fitness sensitivity”, the perception and biased movement along a gradient has been well studied, analogously, in the movement literature as sampling (the authors note this in their discussion on information) and memory. In particular, it has been noted that memory processes analogous to this paper’s fitness sensitivity may limit the spread of an individual governed by an underlying random walk, allowing a home range to form and the forager to be most efficient in depletable heterogeneous landscapes.

We agree with reviewer 1 that our work has many possible links to other areas of the movement ecology literature. As suggested, we have added some text and references to the Introduction that address Individual-based models (see Line 48) and habitat sampling (Line 61 in the revised version with changes tracked).

3. While I realize that the authors can’t make all the connections to this vast literature, I think the ability of the authors to speculate out to a more complex landscape (consisting of more than 2 patches) would increase the readership and scope of the manuscript. In fact, they allude to this on line 285, and I want to hear how they thought this would work.

Thank you for this question. Considering more complex landscapes would make the model more representative of natural habitat structures (e.g., environment having more than 2 patch types and diverse environmental conditions). However, it would not change the assumptions or format of our model beyond introducing more equations and making the overall presentation more challenging. We expect to see qualitatively similar results from a more complex model – more mobile consumers would still be better able to response to variation in fitness and more effectively move to high-fitness patches. We have added a sentence stating this on Lines 307 in the revised manuscript with changes tracked.

4. Like many models of this nature, we often assume that there is no cost with movement and movement occurs instantaneously. Would the authors speculate as to what may happen if this

was included? We have numerous model examples how the marginal value theorem impacts the IFD, could this model add to that understanding?

We agree that including cost during movement would change the results of our model (as it also affects predictions if incorporated into the Ideal Free Distribution model). We suspect that the incorporation of movement costs will depend on the presence or absence of demography (having a larger effect in the absence of demography), and while such an analysis is interesting, we feel that this is beyond the scope of the current paper, even though such costs are unlikely to alter the qualitative results of our two-patch model. We have revised the text on Line 308-312 in the tracked version to reflect this point.

5. The literature cited section has numerous instances with the full journal names are not capitalized and a few instances of title vs. sentence case being used for the article title. I suspect this ought to be standardized.

We made the appropriate changes.

6. The x-axis label for figure 1 & 5 with mobility or fitness sensitivity isn't clear. I also wonder about the y-axis of "density of", as the patches have no dimension and are identical (except in quality) isn't this just number of consumers and resources?

We have changed the x-axis label of Figure 1 & 5 to "Mobility" and edited the figure legend to clarify that movement was determined by both mobility and fitness.

We agree with the referee that the patches in our model have no dimension, but we kept "the density of" as y-axis label for three reasons: 1) in our application density and abundance are interchangeable; 2) many parameters in the model, such as attack rate, are usually measured with respect to density (not abundance); and 3) using "density" implies that our results are independent of patch size, whereas results based on abundance would vary with patch size (because it is not the variable driving the observed results).

7. Line 63 – this sentence is long and could be cleaned up a bit to make the idea clearer.

We have split this sentence into two parts. See Line 75 in the revised version with changes tracked.

Comments from the Referee 2:

1. [...] Overall this paper is well written, and the research is interesting, a nice contribution to the understanding of spatial dynamics and the field of movement ecology.

We appreciate Referee 2's support for the significance of our work.

2. *First of all, I would advice to broaden the introduction by including the term habitat choice. This might be a synonym to fitness sensitivity and an often used term in the field of movement ecology. Still, it is nowhere mentioned in the paper at the moment.*

We thank the reviewer for pointing this out and have added several mentions of habitat choice and of its synonym, habitat selection (see Line 25-26, 33 in the revised version with changes tracked).

3. *Additionally, it would be better to not include a formula in the introduction. Moreover, the reference list should be extended with papers studying habitat choice and/or IFD in IBM models. There are many researchers applying IBM's to investigate movement ecology and you can appeal them by also referring to IBM studies (e.g. Mortier F., Jacob S., Vandegehuchte M.L. & Bonte D. (2019) Habitat choice stabilizes metapopulation dynamics by enabling ecological specialisation. Oikos.).*

We have removed formulas from the introduction and added the suggested citation about IBM studies (Line 48 in the revised version with changes tracked) – see also our response 2 to Reviewer 1.

4. *Clearly state the main features or a clear definition of an Ideal Free Distribution in the introduction. For some readers this might be general knowledge. But for others, this might not be the case.*

We have clarified the definition of Ideal Free Distribution (see Line 29-30 in the revised version with changes tracked).

5. *Not refer to 'demography' but 'consumer demography' instead. Demography can refer to both the resource and the consumer whereas you only want to refer to consumer demography. Demography of the resource is always included in all simulations. (lines 241, 242, 253, 292...)*

We have replaced “demography” with “consumer demography” as suggested. See Lines 173, 183, 230, 253, 254, 257, 266 and 318 in the revised version with changes tracked.

6. *When discussing a possible positive correlation between fitness sensitivity and movement, it might be useful to mention the possible indirect link with body size. According to allometric theory, body size is positively linked to many features of an individual (among which movement speed and perceptual range). You could extend your discussion by mentioning this (reference: Peters 1983).*

Thank you for this suggestion. We have re-written parts of the Discussion to discuss the positive correlations between body size, movement capacities and fitness sensitivity/perceptual capacities (Line 280-283 in the revised version with changes tracked). We now also cite the reference you suggested.

7. *Minor remarks:*

- *Line 30. add a reference*

We have added the reference “Fagan et al. 2017” now on Line 35 as [13] in the revised version with changes tracked.

- *Line 79: ‘a two-patch landscape’ instead of ‘a landscape that is patchy’*

Done.

- *Line 130: change ‘We therefore focus our subsequent analyses’ to ‘Therefore, extra analyses were conducted to’*

Done.

- *Line 133: Beta and lambda are correlated in this section? Clearly state this.*

In this section, the equilibria do not depend on β , so results are valid for both situations (β and λ positively correlated, or λ varies alone). To clarify this, we modified the text on Line 141-144 in the revised version with changes tracked.

- *Line 179: add a reference*

We have added the reference “Mittelbach and McGill 2019” on Line 191 as [36] in the revised version with changes tracked.

- *Line 243: ‘through a possible correlation with fitness’ instead of ‘through fitness’*

Done.

- *Line 279: Shorten the sentence which starts here, it is too long.*

We have rephrased this sentence (now on Line 300-305 in the revised version with changes tracked).

- *Line 284: I would refer to temporal variation instead of environmental change as the former is more general.*

We have modified the text here (Line 306 in the revised version with changes tracked) to refer to temporal variation, with environmental change as one form of temporal variation.

Appendix B

Dear Professor Thomas,

We thank you again for the provisional acceptance of our manuscript "Mobility and its sensitivity to fitness differences determine consumer-resource distributions" for publication in *Royal Society Open Science*. We also thank you for your further advice about comment #4 from Reviewer 1. We have revised our manuscript accordingly and respond in detail below. All co-authors have reviewed and approved the revised manuscript.

Sincerely,

Jing Jiao, on behalf of all authors.

Research Associate
NIMBioS
University of Tennessee

Associate Editor Comments to Author (Professor Len Thomas):

Thank-you for your re-submission, and for taking account of the reviewers' comments. The revisions are largely acceptable, and I am on the point of accepting the paper; however I do not believe you have dealt with Reviewer 1 comment 4 adequately. For one thing, your edited text (lines 287-290 without tracked changes, sentence starting "Our model...") does not form a grammatically correct or comprehensible (to me) sentence. Secondly, the reviewer asked you to speculate as to what would happen if you included a cost to movement and movement taking non-negligible time. You addressed the former obliquely and the latter not at all. Please can you make more of an effort to engage fully with relaxation of both assumptions and provide justification for your speculations.

Thank you. We have reworded the sentence you referred to. It now reads "Our model could also be extended to directly include temporal variation, the timescale of environmental change, and the magnitude of spatial heterogeneities, each of which may interact with animal movement -to affect consumer-resource patterns." (now on Line 287-290 in the revised version without tracked changes).

We have also added speculations about including a movement delay to our model (on Line 295-298). Although we do not expect the addition of costs to qualitatively affect our results,

we suspect that movement delays could lead to cyclical population dynamics, as suggested by Abrams (1992). However, due to the positive correlation between mobility and fitness sensitivity, we still expect that consumers with higher mobility would better sense and forage in high-quality patch(es). We also expect that consumer demography would lead to ideal free-like patterns because this result does not require movement (see Fig. 5). Therefore, we expect that consumer demography would lessen the effects of movement cost and delay. See Line 290-302 for the detailed explanation and citations.

Citation:

Abrams, P.A. 1992 Adaptive foraging by predators as a cause of predator-prey cycles. *Evolutionary Ecology* **6**, 56-72.